# Recent Advances in *Kaempferia* Phytochemistry and Biological Activity: A Comprehensive Review

**DOI:** 10.3390/nu11102396

**Published:** 2019-10-07

**Authors:** Abdelsamed I. Elshamy, Tarik A. Mohamed, Ahmed F. Essa, Ahmed M. Abd-El Gawad, Ali S. Alqahtani, Abdelaaty A. Shahat, Tatsuro Yoneyama, Abdel Razik H. Farrag, Masaaki Noji, Hesham R. El-Seedi, Akemi Umeyama, Paul W. Paré, Mohamed-Elamir F. Hegazy

**Affiliations:** 1Faculty of Pharmaceutical Sciences, Tokushima Bunri University, Yamashiro-cho, Tokushima 770-8514, Japan; elshamynrc@yahoo.com (A.I.E.); yoneyama@ph.bunri-u.ac.jp (T.Y.); mnoji@ph.bunri-u.ac.jp (M.N.); umeyama@ph.bunri-u.ac.jp (A.U.); 2Chemistry of Natural Compounds Department, National Research Centre, 33 El Bohouth St., Dokki, Giza 12622, Egypt; ahmedfathyessa551@gmail.com; 3Chemistry of Medicinal Plants Department, National Research Centre, 33 El-Bohouth St., Dokki, Giza 12622, Egypt; tarik.nrc83@yahoo.com (T.A.M.); ashahat@ksu.edu.sa (A.A.S.); 4Department of Botany, Faculty of Science, Mansoura University, Mansoura 35516, Egypt; dgawad84@mans.edu.eg; 5Plant Production Department, College of Food & Agriculture Sciences, King Saud University, Riyadh 11451, Saudi Arabia; 6Pharmacognosy Department, College of Pharmacy, King Saud University, P.O. Box 2457, Riyadh 11451, Saudi Arabia; 7Pathology Department; National Research Centre, Dokki, Giza 12622, Egypt; abdelrazik2000@yahoo.com; 8Pharmacognosy, Department of Medicinal Chemistry, Uppsala University, Box 574, SE-75 123 Uppsala, Sweden; hesham.el-seedi@ilk.uu.se; 9Department of Chemistry, Faculty of Science, Menoufia University, Shebin El-Kom 32512, Egypt; 10College of Food and Biological Engineering, Jiangsu University, Zhenjiang 212013, China; 11Department of Chemistry & Biochemistry, Texas Tech University, Lubbock, TX 79409, USA; paul.pare@ttu.edu; 12Department of Pharmaceutical Biology, Institute of Pharmacy and Biochemistry, University of Mainz, Staudinger Weg 5, 55128 Mainz, Germany

**Keywords:** *Kaempferia*, traditional medicine, diterpenoids, flavonoids, phenolic, biosynthesis

## Abstract

Background: Plants belonging to the genus *Kaempferia* (family: Zingiberaceae) are distributed in Asia, especially in the southeast region, and Thailand. They have been widely used in traditional medicines to cure metabolic disorders, inflammation, urinary tract infections, fevers, coughs, hypertension, erectile dysfunction, abdominal and gastrointestinal ailments, asthma, wounds, rheumatism, epilepsy, and skin diseases. Objective: Herein, we reported a comprehensive review, including the traditional applications, biological and pharmacological advances, and phytochemical constituents of *Kaempheria* species from 1972 up to early 2019. Materials and methods: All the information and reported studies concerning *Kaempheria* plants were summarized from library and digital databases (e.g., Google Scholar, Sci-finder, PubMed, Springer, Elsevier, MDPI, Web of Science, etc.). The correlation between the *Kaempheria* species was evaluated via principal component analysis (PCA) and agglomerative hierarchical clustering (AHC), based on the main chemical classes of compounds. Results: Approximately 141 chemical constituents have been isolated and reported from *Kaempferia* species, such as isopimarane, abietane, labdane and clerodane diterpenoids, flavonoids, phenolic acids, phenyl-heptanoids, curcuminoids, tetrahydropyrano-phenolic, and steroids. A probable biosynthesis pathway for the isopimaradiene skeleton is illustrated. In addition, 15 main documented components of volatile oils of *Kaempheria* were summarized. Biological activities including anticancer, anti-inflammatory, antimicrobial, anticholinesterase, antioxidant, anti-obesity-induced dermatopathy, wound healing, neuroprotective, anti-allergenic, and anti-nociceptive were demonstrated. Conclusions: Up to date, significant advances in phytochemical and pharmacological studies of different *Kaempheria* species have been witnessed. So, the traditional uses of these plants have been clarified *via* modern *in vitro* and *in vivo* biological studies. In addition, these traditional uses and reported biological results could be correlated *via* the chemical characterization of these plants. All these data will support the biologists in the elucidation of the biological mechanisms of these plants.

## 1. Introduction

From the first known civilization, medicinal plants have met primary care and health needs around the world [1,2,3]. Natural products, derived from plants, have enriched the pharmaceutical industry since time immemorial. So far, people of the developing countries depend upon the traditional medicines to cure daily aliments [4]. The medicinal plants are characterized by a diversity of chemical and pharmacological constituents, owing to their complicity and the abundance of secondary metabolites. There are several factors that caused the variations of the secondary metabolites such as ecological zones, weather, climates, and other natural factors *via* the effects on the biosynthetic pathways [1,2,3].

Zingiberaceae (the ginger family) is distributed worldwide comprising 52 genera and more than 1300 plant species [5,6]. *Kaempferia* is a diverse family with members distributed widely throughout Southeast Asia and Thailand, including some 60 species [5]. Several *Kaempferia* species are used widely in folk medicine, including *K. parviflora*, *K. pulchra*, and *K. galanga*, (Figure 1). In Laos and Thai, traditional medicines derived from *K. parviflora* rhizomes are reported for the treatment of inflammation, hypertension, erectile dysfunction, abdominal ailments [6,7], and improvement of the vitality and blood flow [8]. Japanese use the extract of *K. parviflora* as a food supplement and for the treatment of metabolic disorders [9]. *K. pulchra* is used extensively as a carminative, diuretic, deodorant, and euglycemic, as well as for the treatment of urinary tract infections, fevers, and coughs [4]. The rhizomes of *K. galanga* are used as an anti-tussive, expectorant, anti-pyretic, diuretic, anabolic, and carminative, as well as for the curing of gastrointestinal ailments, asthma, wounds, rheumatism, epilepsy, and skin diseases [10].

Extracts and purified compounds from select *Kaempferia* species are used for the treatment of knee osteoarthritis and the inhibition of a breast cancer resistance protein (BCRP), anti-inflammatory, anti-acne, anticholinesterase, anti-obesity-induced dermatopathy, wound healing, anti-drug resistant strains of *Mycobacterium tuberculosis*, neuroprotective, anti-nociceptive, human immunodeficiency virus type-1 (HIV-1) inhibitory activity, in vitro anti-allergenic, and larvicidal activity against *Aedes aegypti* [4,6,7,8,9,10,11]. The scientific literature such as, Google Scholar, Scifinder, PubMed, Springer, Elsevier, Wiley, Web of Science, were screened in the period between 1972–2019 in order to collect the up-to-date information of the traditional uses/applications, biological studies, and chemical characterization of *Kaempheria* species. All these collected data were addressed and summarized in our review article to highlight the potential ethnopharmacological importance of these plants.

## 2. Materials and Methods

The scientific literature such as Google Scholar, Scifinder, PubMed, Springer, Elsevier, Wiley, Web of Science, etc., including all the traditional uses/applications, biological studies, and chemical characterization of *Kaempheria* species were collected between 1972–2019. All these collected data were adjusted and summarized in our review article due to the potential ethnopharmacological importance of these plants. 

The correlation between the *Kaempheria* species was evaluated based on the main chemical classes of compounds. The data matrix of seven *Kaempferia* species (*K. angustifolia*, *K. elegans*, *K. galanga*, *K. marginata*, *K. parviflora*, *K. pulchra*, and *K. roscoeana*) and six chemical classes (abietanes, labdanes and clerodanes, flavonoids, phenolic compounds, and chalcones) were subjected to principal component analysis (PCA) to identify correlation between different *Kaempferia* species. In addition, the similarity based on the Pearson correlation coefficient was determined *via* subjecting the dataset to an agglomerative hierarchical cluster (AHC). The PCA and AHC were performed using an XLSTAT statistical computer software package (version 2018, Addinsoft, NY, USA, www.xlstat.com).

## 3. Distribution

Zingiberaceae (the ginger family) comprises 52 genera and more than 1300 plant species. *Kaempferia* is distributed worldwide with diverse members occurring throughout southeast tropical Asian countries such as Indonesia, India, Malaysia, Myanmar, Cambodia, and China, as well as Thailand, including some 60 species [5]. *K. pulchra* is a perennial herbal plant and widely cultivated in numerous tropical countries, involving Indonesia, Malaysia, Myanmar, and Thailand [12].

## 4. Traditional Uses

Several *Kaempferia* species are used widely in folk medicine, including *K. parviflora*, *K. pulchra*, and *K. galanga* (Figure 1). In Laos and Thai, traditional medicines derived from *K. parviflora* rhizomes are reported for the treatment of inflammation, hypertension, erectile dysfunction, abdominal ailments [6,7], and improvement of the vitality and blood flow [8]. Japanese folk medicine documented a positive effect of *K. parviflora* extract when used as a food supplement and for the treatment of metabolic disorders [9]. *K. pulchra* is used extensively as a carminative, diuretic, deodorant, and euglycemic, as well as for the treatment of urinary tract infections, fevers, and coughs [4]. *K. galanga* is sold as an industrial crop in the market, and its rhizome has been used as a flavor spice of various cooking [13]. The rhizomes of *K. galanga* is used as an anti-tussive, expectorant, anti-pyretic, diuretic, anabolic, carminative, as well as for curing of gastrointestinal ailments, asthma, wounds, rheumatism, epilepsy, and skin diseases [10]. In Malaysian folk medicines, several gingers belonging to the Zingiberaceae family especially, *Kaempheria* genus, are used in the treatment of several diseases such as stomach ailments, vomiting, cough, bruises, epilepsy, nausea, rheumatism, sore throat, wounds, eyewash, sore eyes, childbirth, liver complaints, muscular pains, ringworm, asthma, fever, malignancies, swelling, and several other disorders [14].

## 5. Biological Activity

Extracts and purified compounds of *Kaempferia* species are used for the treatment of knee osteoarthritis and the inhibition of a breast cancer resistance protein (BCRP), anti-inflammatory, anti-acne, anticholinesterase, anti-obesity-induced dermatopathy, wound healing, anti-drug resistant strains of *Mycobacterium tuberculosis*, neuroprotective, anti-nociceptive, human immunodeficiency virus type-1 (HIV-1) inhibitory activity, *in vitro* anti-allergenic, and larvicidal activity against *Aedes aegypti* [11]. *Kaempheria* plant extracts and isolated compounds demonstrate numerous and promising biological and pharmaceutical activities, which are summarized in Figure 2.

### 5.1. Anticancer Activity

Rhizome ethanolic extracts of *K. galanga* and the purified component ethyl *trans p*-methoxycinnamate (**105**) demonstrate moderate cytotoxic activity against human cholangiocarcinoma (CL-6) cells with IC_50_ of 64.2 and 49.4 μg mL^−1^, respectively. Significant cholangiocarcinoma (CCA) efficacy as indicated by suppressing tumor growth and lung metastasis in CL6-xenografed mice [15] is also observed. Swapana et al. [16] documented that *K. galanga* isopimarene diterpenoids, sandaracopimaradiene-9*α*-ol (**2**), kaempulchraol I (**14**), and kaempulchraol L (**17**) exhibit promising activity against human lung cancer with IC_50_ of 75 µM, 74 µM, and 76 µM, respectively, and mouth squamous cell carcinoma (HSC-2) inhibition with IC_50_ of 70 µM, 53 µM, and 58 µM, respectively [16]. The latter compound, isolated from *K. pulchra*, is reported to have weak anti-proliferative activity against human pancreatic and cervix cancers [17]. Chawengrum et al. [18] stated that *K. pulchra* labdene diterpenoids, (−)-kolavelool (**81**), and (−)-2*β*-hydroxykolavelool (**82**) exhibit cytotoxic activity against human leukemia cells (HL-60) with IC_50_ values of 9.0 ± 0.66 and 9.6 ± 0.88 μg mL^−1^, respectively [18]. Acetone, petroleum ether, chloroform, and MeOH extracts of *K. galanga* rhizomes show moderate cytotoxicity in a brine shrimp lethality bioassay compared with vincristine sulfate as the reference compound [19]. Moreover, a methanolic extract of *K. galanga* rhizomes induces Ehrlich ascites carcinoma (EAC) cell death in a dose-dependent manner [20]. 5,7-Dimethoxyflavone (**86**) isolated from *K. galanga* was found to reduce cancer resistance to tyrosine kinase inhibitors (TKI) by inhibiting breast cancer resistance protein (BCRP), one of the efflux transporters that increased efflux of TKI out of cancer cells. This was observed both in vitro with a dose-dependent increase in the intracellular concentration of sorafenib in MDCK/BCRP1 breast cancer resistance cells, with an EC_50_ of 8.78 μM as well as *in vivo* by increasing sorafenib AUC in mice tissues when co-administered with compound **88**, as reported by kinetic results [21]. The isolated methyl-*β*-*D*-galactopyranoside specific lectin from the rhizome of *K. rotunda* exhibited *in vitro* antitumor activity against Ehrlich ascites carcinoma cells at a pH between 6–9 and a temperature range between 30–80 °C. Tumor inhibition was also observed *in vivo* in EAC-bearing mice [22]. 

The cytotoxicity of MeOH, petroleum ether, and EtOAc extracts against C33A cancer cells *via* MTT and scratch assays compared with essential oils of *K. galanga* rhizomes showed activity for the EtOAc and MeOH fractions at 1000 μg mL^−1^ with 11% and 14% cell viability and weak efficacy with petroleum ether extracted essential oils in a MTT assay. Cell growth inhibition was observed with all extracts in the scratch assay [23]. Compound (**140**) isolated from *K. angustifolia* was described to have strong activity with an IC_50_ of 1.4 µg mL^−1^, which was comparable to 5-fluorouracil as a reference drug. Compound (**138**) also showed moderate inhibition against human lung cancer. 2′-Hydroxy-4,4′,6′-trimethoxychalcone (flavokawain A; **119**) exhibited potent activity against HL-60 and MCF-7 cell lines. The results of Tang et al. [24] revealed that flavokawain A (**119**) exhibited cytotoxic activity against MCF-7 and HT-29 cell lines with GI_50_ values of 17.5 µM (5.5 µg mL^−1^) and 45.3 μM (14.2 µg mL^−1^), respectively. Kaempfolienol (**65**) and zeylenol (**133**) were also found to have moderate activity against HL-60 and MCF-7 cells with IC_50_ values <30 µg mL^−1^ and against HL-60 only with an IC_50_ value of 11.6 µg mL^−1^ respectively [24].

### 5.2. Anti-Obesity Activity

An ethanolic extract, a polymethoxyflavonoid-rich fraction (PMF) and a polymethoxyflavonoid-poor fraction from *K. parviflora* were screened against an obesity-induced dermatopathy system using Tsumura Suzuki obese diabetes (TSOD) mice as an obesity model (Hidaka, Horikawa, Akase, Makihara, Ogami, Tomozawa, Tsubata, Ibuki, and Matsumoto) [11]. The ethanolic extract reduced mouse body weight and the thickness of the subcutaneous fat layer more than the PMF fraction that is used as a dietary supplement in controlling skin disorders caused by obesity [11].

### 5.3. Anti-HIV Activity

Viral protein R (Vpr) is one of the HIV accessory proteins that can be targeted for controlling viral replication and pathogenesis. A CHCl_3_ fraction of *K. pulchra* exhibits Vpr-inhibitory activity at 25l g mL^−1^. In addition, isopimarene type diterpenoids isolated from the rhizomes of the plants, kaempulchraol B (**43**), kaempulchraol D **(45)**, kaempulchraol G (**46**), kaempulchraol Q (**20**), kaempulchraol T (**36**), kaempulchraol U (**50**), and W (**22**) inhibit the expression of Vpr at concentrations from 1.56 to 6.25 µM [25].

### 5.4. Antimicrobial Activity

Arabietatriene (**62**) isolated from *K. roscoeana* exhibits antibacterial activity against Gram-positive bacteria *Staphylococcus epidermidis* and *Bacillus cereus* [26]. Anticopalic acid (**72**), anticopalol (**77**), and 8(17)-labden-15-ol (**68**) isolated from *K. elegans* also exhibited antibacterial activity against *B. cereus* [18]. Acetone, petroleum ether, chloroform, and MeOH extracts of *K. galanga* rhizomes exhibit moderate antibacterial activity against Gram-positive and Gram-negative bacteria in comparison with ciprofloxacin [19]. Ethyl *p*-methoxycinnamate (**105**) also isolated from *K. galanga* rhizomes have been shown based on a resazurin micro-titer assay to inhibit *Mycobacterium tuberculosis* H37Ra, H37Rv, multidrug-resistant, and drug-susceptible isolates with MIC 0.242–0.485 mM [27]. Its essential oil also displays strong antibacterial activity against *Staphylococcus aureus* and *Salmonella typhimurium*, and weak activity against *Escherichia coli* [28]. Moreover, essential oils extracted from three varieties of *K. galanga* exhibited potent larvicidal activity [29]. An ethyl acetate extract of *K. rotunda* inhibits *S. aureus* and *E. coli* [30]. A rhizomes extract of *K. galanga* inhibits Epstein–Barr virus with no cytotoxic effect in Raji cells [14]. In contrast, isolated diterpenoids from *K. roscoeana* exhibited no activity against *Plasmodium falciparum* (Chloroquine-resistant) [26]. Fauziyah et al. [31] described that an ethanolic extract of *K. galanga* alone exhibits 100% growth inhibition of the multi-drug resistant (MDR) *Mycobacterium tuberculosis* (isolates at 500 µg mL^−1^). However, a combination of this extract with streptomycin, ethambutol, and isoniazid showed inhibition values of 55%, 76%, and 50%, respectively. Ethanol, methanol, petroleum ether, chloroform, and aqueous extracts of *K. galanga* rhizome showed antimicrobial activity against human pathogenic bacteria and fungi, while the ethanolic extract exhibited the strongest inhibition of *S. aureus* using an inhibition zone assay [32]. However, flavokawain A (**119**) and other compounds reported from *K. angustifolia* had no antimicrobial activity against tested microbes [24].

### 5.5. Antioxidant Activity

The CHCl_3_ and MeOH extracts of the rhizomes of *K. angustifolia* showed strong antioxidant activity against DPPH expressed with 615.92 mg trolox equivalent (TE)/g of extract. In an azinobis (3-ethyl-benzothiazoline-6-sulfonic acid) (ABTS) assay, MeOH extracts showed good antioxidant properties with a value of 38.87 mg TE/g. However, *n*-hexane extract exhibited significant antioxidant activity with 901.76 mg TE/g in a cupric-reducing antioxidant capacity assay, while EtOAc extract exhibited significant reduction ability against ferric reducing antioxidant power (FRAP) with a value of 342.23 mg TE/g. Also, kaempfolienol (**65**) showed potent free radical scavenging activity in a DPPH assay, as well as, 2′-hydroxy-4,4′,6′-trimethoxychalcone (**119**) in ABTS, CUPRAC, and FRAP assays [33,34]. A methanol extract of rhizomes of *K. galanga* exhibited a concentration-dependent antioxidant activity in DPPH, ABTS, and nitric oxide (NO) radical scavenging assays [20]. Moreover, the essential oil extracts of conventionally propagated and in vitro propagated *K. galanga* had significant DPPH radical scavenging activity [35]. As well, the ethanol extract of *K. rotunda* exhibited antioxidant activity in a DPPH assay with IC_50_ (67.95 μg mL^−1^) [30].

### 5.6. Anti-Inflammatory Activity 

The cyclohexane, chloroform, and ethyl acetate extracts with diarylheptanoids isolated from *K. galanga* showed a pronounced inhibition of Lipopolysaccharides (LPS)-induced nitric oxide in macrophage RAW 264.7 cells compared with indomethacin [13]. The EtOH extract and compounds (**1**, **52**, **53**, **119**, **120**) isolated from *K. marginata* had promising anti-inflammatory activity based on the suppression of NO production and inducible nitric oxide synthase (iNOS) mRNA and cyclooxygenase-2 (COX-2) genes expression [36,37]. Diterpenoids (**9**–**10**) isolated from *K. pulchra* had topical anti-inflammatory activity in 12-*O*-tetradecanoylphorbol-13-acetate-induced ear edema in rats with ID_50_ 330 and 50 µg/ear, respectively. Biological activity may be due to the activation of Maxi-K channels in neurons and smooth muscles [38]. The ethanol extract of *K. parviflora* exhibited potent inhibition of PGE2. The plant extract and 3′,4′,5,7-tetramethoxyflavone (**86**) were also reported to exhibit a dose-dependent inhibition of iNOS-mRNA expression. Additionally, H_2_O, EtOH, EtOAC, CHCl_3_, and *n*-hexane soluble sub-fractions exhibited good in vivo anti-inflammatory activity by decreasing rat paw edema [39]. An 80% EtOH extract reduced UV-induced COX-2 expression in mice skin that was attributed to the anti-oxidative activity of polyphenolics against the oxidizing properties of UV radiation [40]. A 60% EtOH and EtOAc-soluble fraction of 100% methanol extracts of *K. parviflora* decreased knee osteoarthritis, which was likely due to methoxylated flavones [41]. Ethyl *p*-methoxycinnamate (**105**) isolated from *K. galana* inhibited cytokines as IL-1 and TNF*α* and endothelial function in rats [42]. 

Tewtrakul, et al. [43] found that the isolated methoxylated flavonoids from *K. parviflora*, 5-hydroxy-3,7,3′,4′-tetramethoxyflavone (**96**), 5-hydroxy-7,4′-dimethoxyflavone (**93**), and 5-hydroxy-3,7,4′-trimethoxyflavone (**95**) exhibited anti-inflammatory activity against the PGE_2_ production, with IC_50_ values of 16.1 μM, 24.5 μM, and 30.6 μM, respectively [43]. Tewtrakul and Subhadhirasakul [44] described methoxyflavones **96**, **93**, and **95** from a hexane extract of *K.*
*parviflora* rhizomes that exhibited activity against NO release in RAW_264.7_ cells with IC_50_ values of 16.1 μM, 24.5 μM, and 30.6 μM, respectively. In addition, 5-hydroxy-3,7,3′,4′-tetramethoxyflavone (**96**) inhibited PGE_2_ release with an IC_50_ value of 16.3 μM, with negative activity on Tumor Necrosis Factor alpha (TNF-*α*) with IC_50_ >100 μM [44]. Petroleum ether extract from *K. galanga* was active against acute inflammation at 300 mg/kg in rats and inhibited the inflammation and MPO levels at 100 mg kg^−1^ in the chronic model [45].

### 5.7. Anticholinesterase Activity

According to Sawasdee et al. [46], a MeOH extract as well as compounds (**86**–**87**) isolated from *K. parviflora* rhizomes inhibited acetylcholinesterase (AChE) and butyrylcholinesterase (BChE) with greater cholinesterase inhibitory toward AChE and BChE for (**86**), which was an observation of significance in the treatment of Alzheimer’s disease [46]. 

### 5.8. Anti-Mutagenicity Activity

CH_2_Cl_2_ and EtOAc soluble fractions of *K. parviflora* showed anti-mutagenicity and *α*-glucosidase inhibitory activity. Isolated methoxylated compounds (**86**, **97**, **84**, and **92**) from these extracts exhibited potent activity with IC_50_ values of 0.40, 0.40, 0.42, and 0.47 nmol/plate, respectively. Compounds (**88**, **87**, and **91**), also showed significant activity with IC_50_ values of 20.4 μM, 54.3 μM, and 64.3 μM, respectively [47].

### 5.9. Effect on Cytochromes CYP 450

The results listed by Ochiai et al. [48] stated that the continued ingestion of (**88**) isolated from *K. parviflora* decreases liver CYP3A expression, which in turn increased levels of compounds metabolized by CYP3As such as midazolam [48].

### 5.10. Vascular Activity

The oral administration of CH_2_Cl_2_ extract of *K. parviflora* in middle-aged rats was found to decrease vascular responses to phenylephrine, increase acetylcholine-induced vasorelaxation and the production of nitric oxide (NO) from blood vessels, and decrease visceral, subcutaneous fat, fasting serum glucose, triglyceride, and liver lipid accumulation [49]. The effect of intravenous administration of a CH_2_Cl_2_ extract of *K. galanga* to rats reduced the mean arterial blood pressure [50]. This anti-hypertensive effect was attributed to ethyl cinnamate, which is a major compound in the extract [50]. The ethanol extract of rhizomes of *K. parviflora* caused dose-dependent relaxation on aortic rings as well as ileum pre-contracted with phenylephrine and acethylcholine [51].

### 5.11. Adaptogenic Activity

Hexane, chloroform, methanol, and ethanol extracts of *K. parviflora* exhibited adaptogenic activity compared with a crude ginseng root powder used as a reference [52]. A single oral dose of *K. parviflora* rhizome (60% EtOH extract) increased the whole-body potential expenditure in humans [53]. *K. parviflora* was also found to improvement physical fitness and health by decreasing oxidative stress [54].

### 5.12. Xanthine Oxidase Inhibitory Activity

Among the isolated methoxylated flavonoids from *K. parviflora*, (**87** and **86**) inhibit xanthine oxidase activity with IC_50_ values of 0.9 and >4 mM, respectively [9].

### 5.13. Allergenic Activity

Isolated polymethoxyflavones from *K. parviflora* (**86**, **97**), in addition to CH_2_Cl_2_, EtOAc, and H_2_O extracts, alleviated type I allergy symptoms through suppressing Rat Basophilic Leukemia cells (RBL-2H3) cell degranulation, with (**92**) and (**94**) showing the highest anti-allergenic activity [55].

### 5.14. Neurological Activity

A methanolic extract (95% MeOH) of *K. parviflora* exhibited neuroprotective activity by increasing rat hippocampus serotonin, norepinephrine, and dopamine levels in comparison with a vehicle-treated group [56]. An acetone extract of *K. galanga* rhizomes and leaves also exhibited central nervous system depressant activity [57].

### 5.15. Nociceptive Activity.

A *K. galanga* rhizome extract exhibited anti-nociceptive activity in rats that was stronger than aspirin but weaker than morphine. The efficacy was abolished by naloxone, suggesting that the analgesic effect may be centrally and peripherally mediated [58].

### 5.16. Wound-Healing Activity

The co-administration of a *K. galanga* rhizomes extract (95% EtOH) with dexamethazone was found to have wound-healing activity in mice comparable to dexamethazone only [59].

### 5.17. Effects on Sexual Performance

Several 7-methoxyflavones (**86**, **87**, **89**, **91**, **93–95**) isolated from *K. parviflora* rhizomes improved sexual activity in males through the inhibition of PDE5, with **86** being the most potent [60]. The activity was attributed to methoxyls present at positions C5 and C7 [60]. *K. parviflora* rhizome extracts, standardized to 5% DMF, also improve erectile function in healthy men [61]. A *K. parviflora* extract as well as 5,7-dimethoxyflavones augment testosterone production, which decreases age-related diseases and hypogonadism [62]. Improved testosterone levels, sperm count, and sexual performance was observed in streptozotocin (STZ)-induced diabetic rats when treated with a *K. parviflora* extract (aqueous with 1% Tween-80) [63].

### 5.18. Miscellaneous

The rhizome extract (95% ethanolic) of *K. parviflora* reduced obesity via the inhibition of adipogenesis, lipogenesis, and muscle atrophy in mice [64]. In contrast, the *K. parviflora* derivatives of 5-hydroxy-7-methoxyflavone induce skeletal muscle hypertrophy [65]. A *K. parviflora* extract (95% EtOH) served as a potential anti-acne agent with anti-inflammatory, sebostatic, and anti-propioni bacteria activity [66]. 

Recently, *K. parviflora* alcoholic extract at 3–30 µg mL^−1^ was evaluated regarding the molecular mechanisms associated with rheumatoid arthritis for up to 72 h compared with the dexamethasone as positive control [67]. They documented that the EtOH extract significantly decreased the gene expression levels of pro-inflammatory cytokines, inflammatory mediators, and matrix-degraded enzymes, but neither induced apoptosis nor altered the cell cycle. They also reported that the alcoholic extract inhibits cell migration, reduces the mRNA expression of cadherin-11, and selectively reduces the phosphorylation of mitogen-activated protein kinases (P38, MAPKs), signal transducers, and activators of transcription 1 (STAT1) and 3 (STAT3) signaling molecules, without interfering with the NF-κB pathway [67]. 

A *K. galanga* extract (acetone, petroleum ether, chloroform, or methanolic) exhibited dose-dependent anthelmintic activity with strong paralytic activity within one hour and death within 80 min at a 25 mg mL^−1^ concentration [68].

## 6. Chemical Metabolites of *Kaempferia* Species

Chemical profiles of *Kaempferia* exhibited the presence of different types of secondary metabolites such as terpenoids, especially isopimarane phenolic compounds, diarylheptanoids [13], flavonoids [69,70,71], and essential oils [72,73]. This review summarized the reported variety of compound types, including isopimarane, abietane, labdan, and clerodane diterpenoids, flavonoids, phenolic acids, phenyl-heptanoids, curcuminoids, tetrahydropyrano-phenolic, and steroids. Diterpenoids, especially isopimarane types, were the most reported compounds from the plants of this genus, in addition to phenolics, flavonoids, and essential oils. Each class will be described and listed in the following items, and the structures will be summarized in Table 1, Table 2 and Table 3.

### 6.1. Diterpenoids

*Kaempferia* plants were characterized with a predominance of diterpenoids, especially the isopimaranes in addition to abietane, labdane, and clerodane types (Table 1).

#### 6.1.1. Isopimarane-Type Diterpenoids

The isopimaranes reported from the *Kaempheria* species (Table 1) are characterized with the presence of two double bonds; one is mostly ∆^15(16)^, while the other is between ∆^8(9)^ or ∆^8(14)^ [4,25,74]. From the rhizomes of *K. galanga*, 12 usual isopimarenes (**1–8**, **10**, **11**, **14**, and **17**) were observed that contained a ∆^8(14),15^ motif in addition to the rarely reported oxygenated seco-isopimarane (**56**) [16]. From the rhizomes of *K. marginata*, five isopimarenes with a ∆^8(14),15^ motif were observed (**1**, **2**, **52**–**54**) [36]. Only one thumbing isopimarenes, roscorane A (**57**), was reported from *K. roscoeana*, which was characterized by only one double bond ∆^8(9)^ and (7-8)-epoxy, as well as the absence of the exomethylene ∆^15(16)^ [26].

##### Biosynthesis of Isopimarane-Type Diterpenoids

Isopimarane diterpenoids are the most characteristic compounds for *Kaempheria* plants. (*E*,*E*,*E*)-Geranylgeranyl diphosphate (GGPP) is a well-known biosynthesized intermediate of diterpenoids as described by [80]. GGPP is firstly cyclized *via* copalyl diphosphate (CPP) synthases (CPS), and then by the unknown enzyme (PS), affording a charged intermediate (INM). Then, this intermediate is completely cyclized by the enzymatic reactions via the bifunctional (iso) pimaradiene synthases (AoCPS-PS, NfCPS-PS, and AfCPS-PS) (Scheme 1), as described by Xu et al. [81].

#### 6.1.2. Abietane-Type Diterpenoids

Seven abietanes (**58–64**) (Table 1) have been isolated and characterized from the rhizomes of *K. roscoeana*, and one (**65**) was isolated and characterized from *K. angustifolia* [26,33,34]. These highly oxygenated metabolites contain one or more double bonds and an absence of exomethylenes, except for roscotane D (**61**), which contains no double bonds.

#### 6.1.3. Labdane and Clerodane Diterpenoids

After isopimarenes, labdane and clerodane represent major diterpenoid classes from the *Kaempheria* species. Nineteen highly oxygenated labdanes and clerodanes (**66**–**86**) have been reported from *Kaempheria* rhizomes (Table 1) [18,26]. From these isolated labdanes, only (12*Z*,14*R*)-labda-8(17),12-dien-14,15,16-triol (**66**) has been isolated from *K. roscoeana* rhizomes. In contrast, several labdane and clerodane types of diterpenoids have been isolated from *K. elegans* and *K. pulchra* rhizomes collected in Thailand.

#### 6.1.4. Flavonoids 

*Kaempheria* species are characterized by rich biological activity due in part to the presence of a diversity of flavonoids (**86**–**105**) and phenolic compounds (**106–137**) (Table 2). *K. parviflora* rhizomes with flavonoid nuclei contain methoxy groups in specific positions (**86**–**97**) [9,55]. Pyrano-flavone, 2”,2”-dimethylpyrano-[5”,6”:8,7]-flavone (**105**), has been isolated from *K. pulchra* rhizomes collected from Thailand [18], and flavanones (**97–99**) have been isolated and identified from *K. parviflora* rhizomes [70,71]. *K. galanga* contains kaempferol and kaempferide (**98**, **99**) [78].

#### 6.1.5. Phenolic Compounds 

From *K. galanga* rhizomes, diarylheptanoid compounds (**116**, **117**, **122–125**) are reported by Yao, Huang, Wang, and He [13]. From *K. marginata* rhizomes, curcuminoid (**121**) was characterized by Kaewkroek, Wattanapiromsakul, Kongsaeree, and Tewtrakul [36]. From *K. galanga*, rhizomes phenolic acids (**106**–**113**) were the major compounds isolated, including methoxylated cinnamic acid derivatives. Two (4-methoxyphenyl)-propanoates (**114–115**) were also isolated from the *K. galanga* rhizomes [13,50]. *S*- and *R*-isomers at C-4 of phenolic glycosides (**135** and **136**) as well as a rare phenolic glycoside (**137**) were observed in *K. previflora* rhizomes [79]. All the phenolic compounds (**106–137**) are summarized in Table 2. 

#### 6.1.6. Steroids and Triterpenes

Steroids represent a minor class of compounds reported from *Kaempheria* species. Only three steroids, *β*-sitosterol (**138**), *β*-sitosterol-*β*-D-glucoside (**139**), and stigmasterol (**140**) (Table 3) have been reported from *K. marginata* rhizomes [36]. Moreover, only one lanostane type triterpene, (24*S*)-24-methyl-lanosta-9(11), 25-dien-3β-ol (**141**), was isolated from *K. angustifolia* [24].

#### 6.1.7. Volatile Oils

*Kaempheria* species were documented as very rich plants with volatile oils such as *K. galanga* [29,73,82,83], *K. angustiflora* [29], and *K. marginata* [29]. The volatile oil of *K. galanga* has been reported as a potential market product in India and over all the world with market values around 600–700 US$/kg on the international market [83]. Phenylpropanoids and/or cinamates were represented as major constituents of volatile oils derived from *Kaempheria* species followed by monoterpenes [29,73,82]. The phenylpropanoid compound, trans-ethyl cinnamate, was documented as a principal component of volatile oils of all the studied *Kaempheria* species up to date with concentrations varied from 16–35% of the total identified [29,73,82,83]. The volatile oils of *Kaempheria* species were reported to have numerous biological activities such as anti-microbial [83], antioxidant [35], nutraceutical [83], nematicidal toxicity [82], and larvicide activities [29]. Table 4 summarized the main components (**142–157**) of the reported volatile oils of *Kaempheria* species.

## 7. Principal Components Analysis (PCA) and Agglomerative Hierarchical Clustering (AHC) for *Kaempferia* Species

To assess the correlation between the various *Kaempferia* species, chemical classes of different compounds were subjected to PCA and AHC (Figure 3). According to the similarity, the analysis showed that we can group the *Kaempferia* species under three groups: the first group comprised *K. galanga*, *K. marginata*, *K. pulchra*, and *K. roscoeana*, and these species are correlated to isopimaranes compounds. The Pearson correlation coefficient (r) between *K. marginata* and *K. pulchra* was the highest with r = 0.938, while between *K. marginata* and *K. roscoeana*, it was 0.771, between *K. roscoeana* and *K. pulchra*, it was 0.766, and between *K. marginata* and *K. galanga*, it was 0.615 (Table 5).

The second group contained *K. angustifolia* and *K. parviflora* (r = 0.833), and this group showed a close correlation to flavonoids and phenolics. However, the *K. elegans* was separated alone, and showed a close relation to labdane and clerodane compounds. The similarities within each group might be ascribed to the genetic relations, as well as the environmental and microclimatic conditions [1,2,3].

In a study of a genetic variation of *Kaempferia* species based on chloroplast DNA [5], *K. marginata* and *K. galanga* were grouped together, which is agreeable with our results (r = 0.615) according to the PCA data of the present study based on the chemical composition. However, in contrast to the data from the PCA, *K. angustifolia* and *K. parviflora* were separated in different groups, but *K. elegans* and *K. parviflora* were grouped together. In another recent study, based on the DNA and morphological characteristics [84], *K. angustifolia* and *K. parviflora* were grouped together in agreement with the chemical variation of the present study.

## 8. Conclusions

*Kaempheria* species are widely used plants in traditional medicine worldwide. All the biological activity data for these plants and their isolated constituents have resulted in numerous leads for medicinal drugs. Mainly, seven rhizomes of *Kaempheria* plants afforded a vast array of diterpenoids, especially the isopimarane type, along with significant bioactive methoxylated flavonoids. From all these documented chemical and biological results, these plants have been and continue to be a promising source for medicinal natural products and food industrial products.

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
