# Peer review of "Recent Advances in Kaempferia Phytochemistry and Biological Activity: A Comprehensive Review"

_nutrients, 2019, doi:10.3390/nu11102396_

Round 1
Reviewer 1 Report
Abdelsamed I. Elshamy and coauthors presented comprehensive review of phytochemistry and several biological activity of Kaempferia species. Used a wide variety of electronic scientific sources covering the years from 1972 to 2019 they described 141 different compounds and various biological activities related to them or extract or fraction from Kaempferia species. I appreciate the amount of work needed to complete this information, but I have some substantive comments regarding this review as followings.
LINE:
55 “The secondary metabolites are directly correlated with the ecological zones…” what is the context of this sentence ?
56 weather, climates and other natural factors [1-3]. What this mean ?
57,58 The sources of such information should be added.
71 Change font size
76 Citation more than one publication[11] is needed to improve such huge different activity.
202 the numbering of individual compounds should be uniformly in brackets in all text
204 not every abbreviations were explained, please carefully follow the text to list all abbreviations appearing in them
211 extract , extract ?
220 repeated sentence
230 what exactly activity was for compounds 83 ?
262 what mean RBL-2H3?
283 what mean STZ
298 p38 MAPK, STAT1, and 298 STAT3 (abbreviations)
318 it is necessary to draw carbons numbering of structure to indicate position C15,C16 ands rest of structural carbons as well, for example “ Structural elucidation of pimarane and isopimarane diterpenoids: The C-13 NMR contribution” , NPC, 2008, 3(3):399-412, Figure 1 from this paper.
Chapter 5.1.7
Please list the most important components of volatile oils. The table (Tab.1) should be supplemented with the main such a compounds and their chemical structures as well, then you can only talk about the complete phytochemistry presentation of second metabolites.
Tab.1
Schemat of chemical structure isoprene I (comp 36-41) for more convenient of reading should be placed in table just behind of compound 35 in position when structure (36-41) are described.
Comp 51 Isopimara-8(14),15-dien-7-one, it should be Isopimara-8(9),15-dien-7-one
Reviewer 2 Report
I think it is an interesting and complete work on the pharmacological potential of several Kaempferia species.
The summary structure it briefly into objective, material and methods, most relevant results and short conclusion.
Complete figure 1 with an illustration of the complete plant.
Include a chapter of Material and Methods, where the search procedure of the bibliography is said, and the statistical methods are cited.
In conclusion, put, that it is mainly a study of seven species, and that the biological activity comes from secondary metabolites from the rhizome.
